# Natural Variation of Fatty Acid Desaturase Gene Affects Linolenic Acid Content and Starch Pasting Viscosity in Rice Grains

**DOI:** 10.3390/ijms231912055

**Published:** 2022-10-10

**Authors:** Liting Zhang, Yu Xia, Yage Dong, Tianyi Xie, Wenqiang Sun, Sibin Yu

**Affiliations:** 1National Key Laboratory of Crop Genetic Improvement, Huazhong Agricultural University, Wuhan 430070, China; 2College of Plant Science and Technology, Huazhong Agricultural University, Wuhan 430070, China; 3College of Life Science and Technology, Huazhong Agricultural University, Wuhan 430070, China; 4Hubei Hongshan Laboratory, Wuhan 430070, China

**Keywords:** whole grain, nutritional quality, unsaturated fatty acids, fatty acid desaturase, genic marker

## Abstract

Rice, as one of the main food crops, provides a vital source of dietary energy for over half the world’s population. The *OsFAD3* gene encodes fatty acid desaturase, catalyzing the conversion of linoleic acid (LA) to alpha-linolenic acid (ALA) in rice. However, the genetic characterization of *OsFAD3* and its role in the conversion of LA to ALA remains elusive. Here, we validated the effects of two homologous genes, *OsFAD3-1* and *OsFAD3-2*, on the ALA and LA/ALA ratio in rice grains using near-isogenic lines. Two major haplotypes of *OsFAD3-1* are identified with different effects on the ALA and LA/ALA ratio in rice germplasm. High expression of *OsFAD3-1* is associated with high ALA accumulation and eating quality of rice grains. Overexpression of *OsFAD3-1* driven by a seed-specific promoter increases the ALA content up to 16-fold in the endosperm. A diagnostic marker is designed based on an 8-bp insertion/deletion in the *OsFAD3-1* promoter, which can recognize *OsFAD3-1* alleles in rice. These results indicate that *OsFAD3-1* is a useful target gene in marker-assisted breeding programs to improve varieties with high ALA and appropriate LA/ALA ratio in brown rice.

## 1. Introduction

Rice (*Oryza sativa* L.) is one of the most important staple foods and provides the source of dietary carbohydrates and energy for billions of people worldwide. It is usually consumed after removing and polishing the outer layers and the embryo, which will cause considerable loss of nutrients, dietary fiber, and beneficial compounds [1,2]. Therefore, whole grain rice (or brown rice) consisting of pericarp, aleurone, and germ, is recommended as a natural healthy diet, due to its nutritional value and potential health benefits in comparison to the milled or white rice [3,4,5].

Brown rice is a rich source of unsaturated fatty acids such as oleic (18:1), linoleic (18:2), and alpha-linolenic (C18:3) acids, which have nutritional and functional significance [6,7]. It has been reported that linoleic acid (LA) and alpha-linolenic acid (ALA) are essential unsaturated fatty acids, as they cannot be synthesized by the human body and need to be solely supplied through a diet [8]. In addition, a skewed ratio of LA to ALA (LA/ALA) in the diet is one of the most prevalent nutritional problems in the world and may explain the increasing incidence of cardiovascular diseases and inflammatory/autoimmune diseases [9,10]. The appropriate low ratio of LA/ALA intake is suggested for reducing the risk of chronic diseases [11]. Rice seeds contain a relatively low ALA content and a high ratio of LA to ALA [12]. Therefore, the increased alpha-linolenic acid content and reduced LA/ALA ratio in brown rice is of importance and will potentially contribute to human health benefits.

Fatty acid desaturase (FAD) is a key enzyme in the biosynthesis of polyunsaturated fatty acids [13]. The characterization of FAD genes from rice offers an abundance of candidates for the production of nutritionally beneficial unsaturated fatty acids. It has been reported that the desaturation of oleic acid to linoleic acid is catalyzed by the enzyme ω-6 fatty acid desaturase 2 (FAD2) and the conversion of LA to ALA is promoted by ω-3 fatty acid desaturase 3 (FAD3) [14,15]. The rice genome encodes four *FAD2* genes. Among them, *OsFAD2-1* is located on rice chromosome 2, and the other three *FAD2* genes are clustered on chromosome 7. *OsFAD2-1* is highly expressed in rice seeds and plays a pivotal role in the conversion of oleic acid to linoleic acid in rice grains [16,17]. RNA interference (RNAi) suppression of the *FAD2* genes resulted in an increase of oleic acid and a decrease of linoleic acid in rice [18]. CRISPR/Cas9-induced *OsFAD2-1* mutants increased the oleic acid more than two times and decreased LA dramatically to an undetectable level in rice grains compared to the wild type [19]. The rice genome harbors four *FAD3* genes encoding ω-3 fatty acid desaturase, i.e., *OsFAD3-1*, *OsFAD3-2*, *OsFAD7,* and *OsFAD8* [20,21]. Of them, *OsFAD3-1* and *OsFAD3-2* are orthologous genes localized on the endoplasmic reticulum (ER) membrane, which catalyze LA to ALA in rice seeds [20]. Overexpression of *OsFAD3-2* led to a 27.9-fold increase of ALA content in the rice seeds, higher than that of the wild type [12]. Moreover, the chloroplast-localized *OsFAD7* and *OsFAD8* had less effect on increasing ALA content in rice seeds [12,22]. Recently, some efforts have been made to explore candidate genes involved in oil metabolism. Four genes (*PAL6*, *LIN6*, *MYR2*, and *ARA6*) have been identified to contribute to natural variation in unsaturated fatty acid compositions in rice germplasm [23]. However, the knowledge of FAD genes and their functions in rice seeds is incomplete. The natural variation of *OsFAD3-1* and its role in the conversion of LA to ALA in rice remains elusive.

In this study, we used the near-isogenic line and its derived segregating population and overexpression transgenic experiment with aims to validate the effect of *OsFAD3-1* on the ALA and LA/ALA ratio in rice seeds and to explore the allelic variation of *OsFAD3* in rice germplasm for designing diagnostic markers. The results can be deployed in genomic breeding programs to develop nutritional rice varieties.

## 2. Results

### 2.1. Effects of OsFAD3-1 and OsFAD3-2 on Unsaturated Fatty Acids

There are two homologous *FAD3* genes (*OsFAD3-1* and *OsFAD3-2*) in rice [20]. Their amino acid sequence similarity is 95% (http://rapdb.dna.afrc.go.jp/, accessed on 22 July 2022). To compare their effects on unsaturated fatty acid compositions, we developed two substitution lines (named IL3.1 and IL3.2) that contain a particular Nipponbare (NIP) allele (*OsFAD3-1*^NIP^ or *OsFAD3-2*^NIP^) within the background of an indica variety Zhengshan97 (ZS97) (Appendix A). The two substitution lines with the introduced NIP alleles show a significant difference in the unsaturated fatty acid content in the embryos from that of ZS97, while they do not differ in the contents of total fatty acids and stearic acids (Figure 1A–C). The two lines both exhibit significant reduction of oleic acid (OA) and ALA contents, and an increase of LA content (Figure 1D–F) compared to ZS97. Consequently, the two lines display a lower ratio of OA/LA ratio and a higher ratio of LA/ALA in the embryos than ZS97 (Figure 1G,H). These data suggest that both *OsFAD3-1* and *OsFAD3-2* catalyze the transformation of LA to ALA. The replacement of the ZS97 alleles by the NIP alleles of *OsFAD3-1* or *OsFAD3-2* caused a reduced ALA and an increased ratio of LA/ALA ratio in the rice embryo.

### 2.2. Validate of Genetic Effect of OsFAD3-1 on the LA/ALA Ratio

To verify the *OsFAD3-1* effect, an IL3.1-derived F_2_ population was generated for further genetic analysis. The population comprised of 110 genotyped individuals, using seven polymorphic markers to target the corresponding three introduced segments, as IL3.1 contains one *OsFAD3-1* region on chromosome 11 and two other introduced NIP segments on chromosomes 5 and 9 (Figure 2A). The fatty acid content in the embryos was evaluated in each individual. QTL analysis reveals that the *OsFAD3-1* region exhibits a major effect on the LA content and LA/ALA ratio, explaining 58.5% of the phenotypic variance of the ratio in the population (Figure 2B). Analysis of variance shows significant differences in the contents of LA, ALA, and the LA/ALA ratio among three genotypes assayed by the genic marker M3. *OsFAD3-1*^NIP^ yields a higher LA content and LA/ALA ratio, but a lower ALA than *OsFAD3-1*^ZS97^. The heterozygote contains LA, ALA, and the LA/ALA ratio similar to that of the *OsFAD3-1*^ZS97^ homozygote (Figure 2C). However, two markers, ID0508 (on Chr5) and ID0904 (on Chr9), show no or a marginal effect on the fatty acid contents and LA/ALA ratio in the embryos (Appendix A). These results indicate that *OsFAD3-1* exhibits incomplete dominance, with the *OsFAD3-1*^ZS97^ allele having a strong effect on the transformation of LA to ALA.

### 2.3. Expression Variation of OsFAD3-1 in Rice Seed

Expression profile analysis reveals that *OsFAD3-1* is expressed in various tissues, particularly at the reproductive stage with a relatively high level in the early seeds (embryos) around 10 days after flowering (Figure 3A; https://ricexpro.dna.affrc.go.jp/, accessed on 23 August 2021). The transcript level of *OsFAD3-1* in the embryo is significantly higher than that of the endosperm during early seed development. qRT-PCR analysis confirms that *OsFAD3-1* is expressed in the spikelets at 3–12 days after flowering, with a higher expression level of *OsFAD3-1*^ZS97^ than *OsFAD3-1*^NIP^ throughout the reproductive stage, as revealed by using near isogenic lines (NILs) that harbor contrasting alleles in the background of ZS97 (Figure 3B). The same pattern with a higher expression level of *OsFAD3-**2*^ZS97^ than *OsFAD3-**2*^NIP^ was also observed (Appendix A). Consistent with the expression difference, total fatty acids dominantly accumulated in the rice embryo, approximately 14-fold more than that in the endosperm (Figure 3C). Moreover, the three fatty unsaturated acids constitute approximately 75% of the total fatty acids in the embryo (Appendix A). The unsaturated fatty acids in the embryo exhibit a significant difference, despite the contents of the total fatty acids and stearic acids do not differ significantly between NIL(*OsFAD3-1*^ZS97^) and NIL(*OsFAD3-1*^NIP^) (Figure 3C–G). NIL(*OsFAD3-1*^ZS97^) contains higher contents of OA and ALA, but lower content of LA than NIL(*OsFAD3-1*^NIP^) (Figure 3H,I). NIL(*OsFAD3-1*^ZS97^) reduces the LA/ALA ratio by 35.7% compared to NIL(*OsFAD3-1*^NIP^). These results demonstrate that highly expressed *OsFAD3-1*^ZS97^ may cause high ALA content and low LA/ALA ratio in rice embryos.

### 2.4. Overexpression of OsFAD3-1 Increases Alpha-Linolenic Acid Content in Rice

To determine the function of *OsFAD3-1* on the conversion of LA to ALA, we generated transgenic plants overexpressing *OsFAD3-1*^ZS97^ driven by a seed-specific promoter of *GluA-1*. Two independent lines (OE1 and OE2) overexpressing *OsFAD3-1* were evaluated in fatty acid content in the seeds. OE1 and OE2 significantly decrease LA and increase ALA in the embryo and endosperm compared to the wild type (WT), leading to a low ratio of LA/ALA (Figure 4). Notably, the overexpression *OsFAD3-1* lines display a significant increase in ALA content up to 16-fold and 10-fold in rice endosperm and embryo, respectively, as compared to the WT. These results indicate that *OsFAD3-1* plays a pivotal role in the conversion of LA to ALA in rice.

### 2.5. Allelic Variation of OsFAD3-1 in Rice Germplasm

A comparison of the sequence variation of *OsFAD3-1* in a panel of 501 rice accessions obtained from the RiceVarMap database (http://ricevarmap.ncpgr.cn, accessed on 29 October 2021) reveals a total of 29 single nucleotide polymorphisms (SNPs) and insertion/deletions (Indels) in the germplasm (Figure 5A). Among them, three variations in the coding region cause amino acid changes, and one 8-bp indel at -193 bp and one 5-bp indel at -51 bp upstream of the start codon (ATG) of the gene may result in the expression variation (Appendix A).

Based on these variations, two major groups or haplotypes are identified in the accessions, of which Hap1 and Hap2 are dominant in representative japonica varieties like NIP and indica varieties like ZS97 (Figure 5), respectively. We further developed a genic marker (FA) based on this 8-bp indel in the gene promoter. To test the specificity and applicability of the marker, other thirty rice accessions are genotyped and phenotyped. The developed marker can classify the thirty rice accessions into two haplotypes (Appendix A). Hap1 exhibits a higher LA content but a lower ALA content than Hap2, while there is no significant difference in stearic acid and oleic acid contents between the two group seeds (Figure 5B–E). Thus, Hap2 contains a relatively higher OA/LA ratio and lower LA/ALA ratio than Hap1 (Figure 5F,G).

### 2.6. OsFAD3-1 Alters the Cooking and Eating Quality of Rice

Starch-pasting viscosity is one of the important physicochemical characteristics of starch related to the eating and cooking quality of rice. To further clarify whether *OsFAD3-1* affects grain quality, we compared pasting viscosity profiles in grains of paired NILs. The data reveal that *OsFAD3-1* alters pasting viscosity values in rice grains (Figure 6A–E), with significant decreases of hold viscosity (HP), final viscosity (CP), setback (SB), and consistence viscosity (CS) values, while a significant increase of breakdown (BD) in NIL(*OsFAD3-1*^ZS97^) as compared to NIL(*OsFAD3-1*^NIP^). Consistently, overexpression of *OsFAD3-1*^ZS97^ decreased HP, CP, SB, and CS, and increased BD (Figure 6F–J). Therefore, *OsFAD3-1* is involved in the pasting viscosity of rice grains.

## 3. Discussion

ALA belongs to the polyunsaturated fatty acids and it is an essential fatty acid in the human diet [24,25]. The genetic characterization of *OsFAD3* conferring the conversion of LA to ALA would be beneficial for improving the nutritional quality of rice grains. As the majority (94%) of fatty acids and most unsaturated fatty acids are mainly present in the embryo (Figure 3C), we have focused on fatty acid content in the embryo of brown rice. Using NILs and overexpression transgenic lines, we found that both *OsFAD3-1* and *OsFAD3-2* have a similar function in the conversion of LA to ALA in rice grains. Moreover, two major types of allelic variation in *OsFAD3-1* identified have different effects on fatty acid conversion in rice germplasm. The transcript level of *OsFAD3-1* due to its promoter variation is closely associated with the ALA and LA/ALA ratio in rice (Figure 5). Highly expressed *OsFAD3-1*^ZS97^ significantly increases ALA, and decreases LA in rice grains, leading to a reduction of the LA/ALA ratio compared to *OsFAD3-1*^NIP^ (Figure 1 and Figure 3). Overexpression of *OsFAD3-1*^ZS97^ results in an approximately 16-fold increase in ALA content in the rice endosperm. These data suggest that *OsFAD3-1*^ZS97^ is a favorable allele for the improvement of the LA/ALA ratio in rice grains to enhance its nutritional value.

The consumption of brown rice is increasing because of its nutritious and health compared to the milled or polished rice [26]. However, the eating quality of brown rice is usually not desirable and needs to be improved. Although fatty acids are minor components of the rice endosperm, they may play an important role in starch quality [27,28,29]. However, the effect of unsaturated fatty acids on the eating quality of rice is unclear. The present results reveal that *OsFAD3-1* influences starch-pasting viscosity values in the endosperm. *OsFAD3-1*^ZS97^ significantly increased breakdown viscosity and reduced setback viscosity compared to *OsFAD3-1*^NIP^ (Figure 6). In line with these results, overexpression of *OsFAD3-1*^ZS97^ positively affects starch pasting properties of rice grains. As the attributes of high BD and low SB are associated with good tasty, softening, and stickiness of cooked rice [30,31], the *OsFAD3-1* alleles with a high expression level are useful to improve the cooking and eating quality of rice. Therefore, *OsFAD3-1* would be the potential target for the improvement of rice quality regarding both the cooking and eating quality and nourishment of brown rice.

Currently, many nutritious components like fatty acids in seeds are measured by a gas chromatography–mass spectrometry instrument (GC–MS) or other chemical methods [32,33,34]. Due to the limitations of high cost and because they are extremely time-consuming, these methods are not applied widely in the breeding of nutritious varieties. In this context, a large effort is required to identify the genes associated with the metabolism of fatty acid compositions and to develop a gene-specific marker system for the improvement of nutritional quality in crops [35]. In the present study, a diagnostic marker developed for *OsFAD3-1* can accurately recognize the corresponding alleles with a rapid and cost-effective advantage over chemical methods. Using the specific marker, we could identify desirable *OsFAD3-1* alleles for high ALA at early plant growth with any other tissue except for seeds. Overall, our data provide the candidate genes and diagnostic markers useful in molecular breeding programs for developing varieties with high ALA and appropriate LA/ALA ratio in rice grains.

## 4. Materials and Methods

### 4.1. Plant Materials

Chromosome segment substitution lines (ILs) each carrying a particular *OsFAD* gene from the variety Nipponbare (NIP) introduced into the indica variety Zhengshan97 (ZS97) were developed using a marker-assisted backcross scheme [36]. The ILs were crossed with ZS97 to generate an F_2_ population and paired near isogenic lines (NILs) for the validation of the gene effect. All the rice materials were grown at the experimental stations of Huazhong Agricultural University in Wuhan (30.4° N, 114.2° E) or Lingshui (18.2° N, 108.9° E).

### 4.2. Genotype and QTL Analyses

Total DNA was extracted following the CTAB method [37]. Primers were designed using Primer 5.0 (http://redb.ncpgr.cn/modules/redbtools/, accessed on 10 June 2018) according to sequence variation. PCR amplification was carried out with a 20 μL system as described previously [38]. The primers used in this study are listed in Appendix A.

The sequence variations of *OsFAD3-1* were used for haplotype analysis in a panel of 501 rice accessions. The genomic sequences of *OsFAD3-1* (LOC_Os11g01340) and *OsFAD3-2* (LOC_Os12g01370) were obtained from the Rice Genome Browser (http://rice.uga.edu/cgi-bin/gbrowse/rice/, accessed on 3 December 2021). The haplotypes were classified according to all SNPs according to the method described [39].

QTL analysis was performed in an F_2_ population using the IciMapping 4.2 software (https://isbreeding.caas.cn/rj/qtllcmapping/, accessed on 21 October 2020). Significant differences were determined among rice accessions or between pairwise lines using one-way analysis of variance (ANOVA) or Student’s *t*-test.

### 4.3. Expression Analysis

Total RNA in the rice spikelets at 3, 5, 8, and 10 days after flowering (DAF) were extracted separately, using the TRIzol Reagent Kit (Aidlab Biotechnologies Co., Beijing, China). The first-strand cDNA was synthesized using a Star Script II First-strand cDNA Synthesis Kit (Promega, Madison, WI, USA). Reverse transcription PCR (qRT-PCR) was performed with SYBR Green Master (Roche Diagnostics, Mannheim, Germany) on an ABI ViiA7 Real-Time PCR System (Applied Biosystems, Foster City, CA, USA) as described previously [40]. The rice *Ubiquitin* gene (LOC_Os03g13170) was used as an internal reference. Gene expression quantity was analyzed as described previously [41]. Three biological replicates and three technical replicates were performed for each sample.

### 4.4. Vector Construction and Transformation

To construct the seed-specific overexpression vector, *OsFAD3-1* and a seed-specific promoter (named Gt13) of *GluA-1* (LOC_Os01g55690) were cloned into linearized *pCAMBIA1301* (CAMBIA) with *Xba*I and *Hind*III successively using the Seamless Assembly and Cloning Kit (Aidlab Co., Beijing, China). *OsFAD3-1* was amplified from the ZS97 cDNA, and the seed-specific promoter *Gt13* was obtained from the japonica rice NIP using specific primers (Gt13 and XFAD3FL1) [42,43]. The construct was transferred into a variety NIP using agrobacterium-mediated transformation. The first-generation (T0) transgenic plants were genotyped by PCR using specific primers (GaF0 and FaR3) and self-pollinated twice to generate homozygous T3 lines for further analysis. All primers used for the transgenic experiment are listed in Appendix A.

### 4.5. Measurement of Fatty Acids

Fatty acids of the embryo and milled endosperm were extracted and measured by a gas chromatography-mass spectrometry instrument (GC-MS) as described previously [23]. Briefly, approximately 20 mg of healthy rice seeds were collected and dehulled using a rice dehuller (JLGL-45) to produce brown rice grains. Two fractions (embryo and endosperm) of brown rice were separated with a razor blade. The embryo section was ground to powder with liquid nitrogen. The endosperm fraction was further milled, and the milled sample was ground into flour using a CT410 Cyclotec mill (FOSS, Hillerod, Denmark). A sample of 0.2 mg embryo flour or 2 g endosperm powder were added with 4.5 mL 5% sulfuric acid methanol solution (containing 0.01% butylated hydroxytoluene) and 0.1 mL heptadecanoic acid (C17:0, 16.2 mol/L), bathed in water at 88 °C for 3 h, then cooled down to room temperature, and added with 2 mL n-hexane and fully swirled, then centrifuged at 3500 rpm for 10 min to obtain supernatant. Quantification was performed in GC-MS (QP2010ULTRA, Shimadzu, Kyoto, Japan) with a chromatographic SH Stabil Wax column. The total fatty acid content was calculated as the sum of all identified fatty acid composition (mg/g) of sample weight. Individual fatty acids were expressed as a percentage (%) of total fatty acids. Each sample was assayed three times, and the mean value was used for QTL analysis and statistical analyses.

### 4.6. Pasting Viscosity Value Analysis

The milled rice was ground into flour and passed through an 80-mesh sieve for the analysis of pasting properties. The starch paste characteristics of the flour samples were evaluated by employing a Rapid Visco Analyser (RVA 4500, Perten Instruments, Stockholm, Sweden) [44]. Six parameters for the pasting viscosity, including peak viscosity (PV), hold viscosity (HP), final viscosity (CP), breakdown (BD = PK − HP), setback (SB = CP − PK), consistence viscosity (CS = HP − CP), were derived from the software, Thermal Cycle for Windows (TCW3) [45].

## 5. Conclusions

*OsFAD3-1* encoding fatty acid desaturase catalyzes the conversion of LA to ALA and affects the eating quality of rice grains. Highly expressed *OsFAD3-1* increases ALA and reduces the LA/ALA ratio in rice grains. A gene-specific marker of *OsFAD3-1* developed can accurately distinguish *OsFAD3-1* alleles. The genetic characterization of *OsFAD3* facilitates a better understanding of unsaturated fatty acid metabolism, which is useful in brown rice breeding programs to improve varieties with high ALA and appropriate LA/ALA ratio for health benefits.

## Figures and Tables

**Figure 1 ijms-23-12055-f001:**
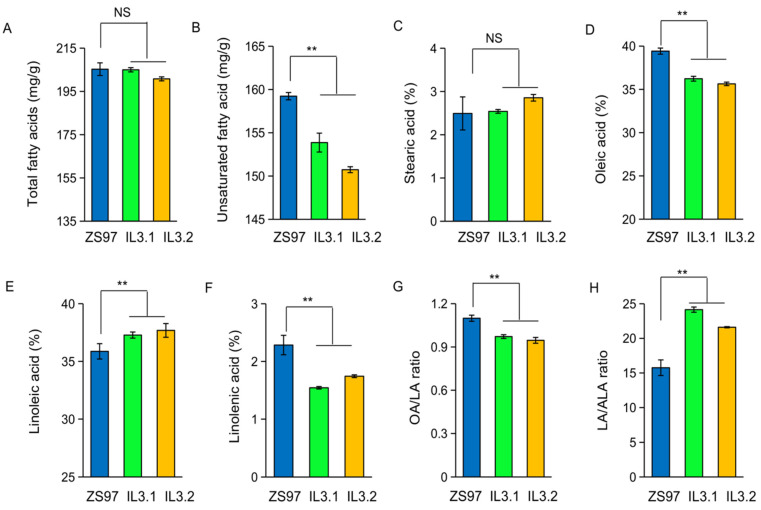
Comparison of fatty acid content in the embryo between ZS97 and two introgression lines containing a particular *OsFAD* gene from Nipponbare, IL3.1 (*OsFAD3-1*^NIP^) and IL3.2 (*OsFAD3-2*^NIP^). (**A**) Total fatty acids (mg/g); (**B**) unsaturated fatty acid (mg/g); (**C**) stearic acid; (**D**) oleic acid (OA); (**E**) linoleic acid (LA); (**F**) alpha-linolenic acid (ALA, C18:3); (**G**) the OA/LA ratio; (**H**) the LA/ALA ratio. Values are means ± SE (*n* = 3 replications). Significance is indicated by *t*-test. ** *p* < 0.01, NS, non-significant.

**Figure 2 ijms-23-12055-f002:**
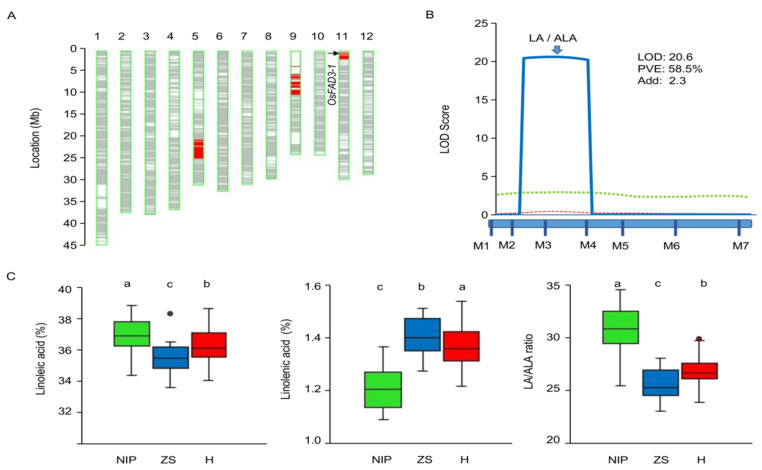
The genetic effect of *OsFAD3-1* on unsaturated fatty acids in a segregated population derived from a cross of ZS97 and the introgression line IL3.1 (*OsFAD3-1*^NIP^). (**A**) Graphic genotype of IL3.1. (**B**) QTL mapping in the segregating population (*n* = 111); blue solid line, LA/ALA ratio; green dotted line, alpha-linolenic acid (ALA); red dotted line, linoleic acid (LA); LOD, logarithm of the likelihood of odds; PVE, phenotypic variation explained; Add, additive effect. (**C**) Differences of LA, ALA, and the LA/ALA ratio in the embryo of three genotypes assayed by the genic markers M3. Black dots in the boxplot represent outliers. NIP, homozygous Nippobare alleles; ZS, homozygous ZS97 alleles; H, heterozygote. Significant differences are indicated as different lowercase letters (a,b,c) by ANOVA.

**Figure 3 ijms-23-12055-f003:**
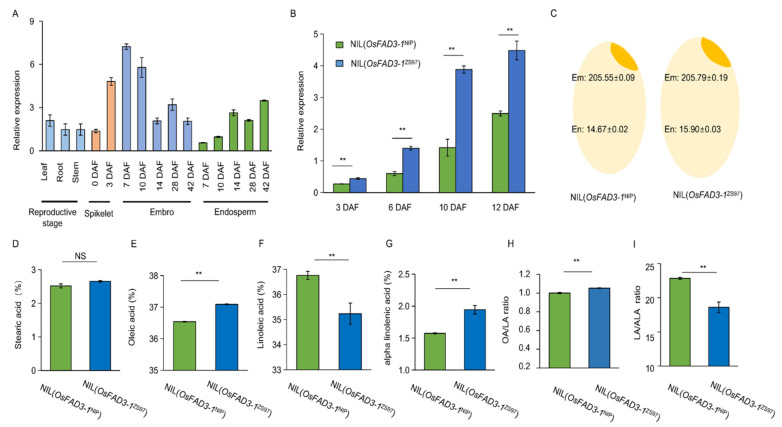
Expression analyses of *OsFAD3-1* and differences in the unsaturated fatty acid content between near-isogenic lines, NIL(*OsFAD3-1*^NIP^) and NIL(*OsFAD3-1*^ZS97^). (**A**) Expression profile of *OsFAD3-1* in various tissues of Nipponbare at the reproductive stage; DAF, days after flowering. (**B**) *OsFAD3-1* expression in the spikelets of 3, 6, 10, and 12 DAF in NILs. The *Ubiquitin* as an internal reference. (**C**) Total of fatty acids (mg/g) in the embryo (Em) and endosperm (En) of NILs. (**D**–**I**) Relative contents of fatty acids in the embryo of the NILs, stearic acid (**D**), oleic acid (**E**), linoleic acid (**F**), alpha-linolenic acid (**G**). OA/LA, linoleic acid/oleic acid ratio (**H**), LA/ALA, linoleic acid/alpha-linolenic acid ratio (**I**). Values are means ± SE, and three replications were performed in each analysis. Significant difference is indicated by *t*-test at ** *p* < 0.01, NS: non-significant.

**Figure 4 ijms-23-12055-f004:**
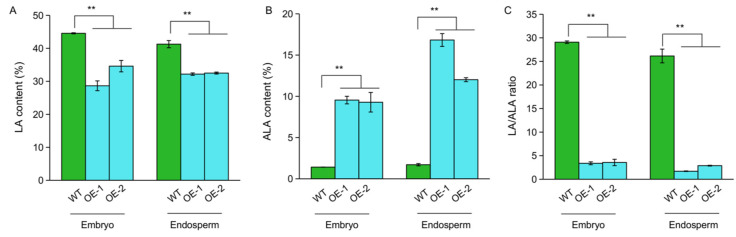
Overexpression (OE) of *OsFAD3-1* in rice increased linolenic acid content. (**A**,**B**) The relative contents (%) of linoleic acid (LA), alpha-linolenic acid (ALA) in the corresponding samples, respectively; (**C**) LA/ALA ratio. Values are means ± SE. ** *p* < 0.01 against wild type (WT) by *t*-test.

**Figure 5 ijms-23-12055-f005:**
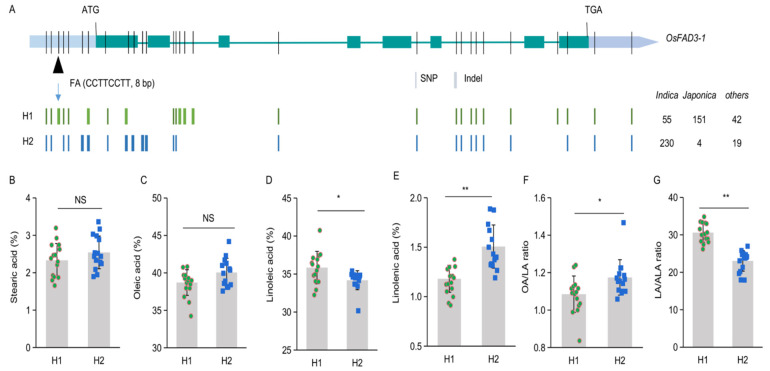
Two major haplotypes of *OsFAD3-1* in rice germplasm (n = 263). (**A**) Schematic diagram of the gene showing relative locations of single nucleotide polymorphisms (SNP) and insertion/deletion (Indel). The relative location of diagnostic marker FA for the 8-bp Indel is indicated by arrows. (**B**–**G**) The average content (%) of stearic acid (**B**), oleic acid (**C**), linoleic acid (**D**), alpha-linolenic acid (**E**); and OA/LA, linoleic acid/oleic acid ratio (**F**), LA/ALA, linoleic acid/alpha-linolenic acid ratio (**G**), in the grains of representative haplotypes, H1 (*n* =15) and H2 (*n* = 15). Values are means ± SD. * *p* < 0.05, ** *p* < 0.01, NS, non-significant.

**Figure 6 ijms-23-12055-f006:**
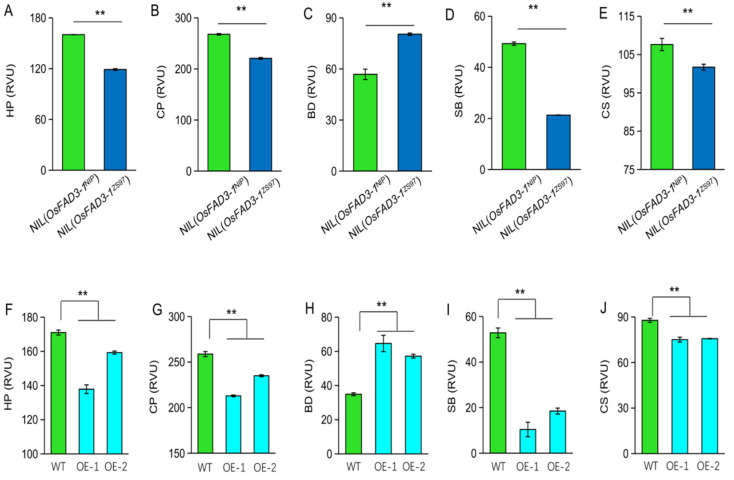
*OsFAD3-1* is involved in starch pasting viscosity in milled rice. (**A**–**E**), differences between NILs in hold viscosity (HP), final viscosity (CP), breakdown (BD), setback (SB), consistence viscosity (CS), respectively; (**F**–**J**) differences in the values among *OsFAD3-1* overexpression lines (OE) and wild type (WT). Values are means ± SE. ** *p* < 0.01 against WT by *t*-test.

## Data Availability

Not applicable.

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
