# Peer review of "Natural Variation of Fatty Acid Desaturase Gene Affects Linolenic Acid Content and Starch Pasting Viscosity in Rice Grains"

_ijms, 2022, doi:10.3390/ijms231912055_

Round 1
Reviewer 1 Report
In this paper, Zhang et al., characterized the genes FAD3 homologous genes in rice. They evaluated their effects on unsaturated fatty acids but a loss and gain of function approaches using substitution and seed specific overexpressing lines, respectively. Moreover, they identified natural variation of FAD3-1 gene within O. Sativa germplasm which can be sorted into two major distinct haplotypes that correlate with unsaturated fatty acids contents. Finally, the authors studied the effect of FAD3-1 on the starch pasting viscosity.
Overall, this article is well written, straightforward and provide further understanding of Linolenic acid biosynthesis in seeds. Well done!
I noticed some minor points that needs to be improved.
L19: define better “good taste quality” this is highly suggestive.
L68: to contribute to
L88 (and where the ratio LA/ALA is presented): It’s not clear to me why the authors assume that alpha linolenic acid levels is equal to linolenic acid levels. Please justify why, and/or provide the content of gamma-linolenic acid or modify accordingly.
Figure 3H: the statistical bar is not well aligned
L116: No data shown
L133: “Than that of” instead of “than that in”
L163: Specify which seed specific promoter is used and add the reference.
L331: “taste” is highly suggestive and thus need to be changed for more accuracy.
Author Response
Thank you for the positive comments. Our responses to your concerns are below.
L19: define better “good taste quality” this is highly suggestive.
We have changed “good taste quality” to “eating quality”.
L68: to contribute to
Done. Thanks.
L88 (and where the ratio LA/ALA is presented): It’s not clear to me why the authors assume that alpha linolenic acid levels is equal to linolenic acid levels. Please justify why, and/or provide the content of gamma-linolenic acid or modify accordingly.
The gamma-linolenic acid content is of trace and could not be detected in the rice embryo by GC-MS. However, to be more accuracy, we have replaced “linolenic acid” with “alpha-linolenic acid” in the revised manuscript.
Figure 3H: the statistical bar is not well aligned
We have updated Figure 3.
L116: No data shown
We have added a supplementary file (Figure S1), showing the one-way analysis of variance for the marker effect.
L133: “Than that of” instead of “than that in”
Done.
L163: Specify which seed specific promoter is used and add the reference.
We have rephrased the sentence with “a seed-specific promoter of GluA-1” in L164. How to obtain the specific promoter of GluA-1 and the references for the promoter are provided in L298-299.
L331: “taste” is highly suggestive and thus need to be changed for more accuracy.
We have changed “taste” to “eating”.
Reviewer 2 Report
Zhang et al. reported their study on the natural variation of OsFAD3 gene affects ALA content and starch pasting viscosity in rice grains. The methods are appropriate, the data and analysis methodology are sound. I have a few concerns about the manuscript: 1) line 83, the authors should provide graphic genotype for IL3.1 and IL3.2; 2) Page 2 line 92-94, there is no data to support this results, pls provide supporting data; 3) Page 4, if possible, the authors could provide the expression data for OsFAD3-2 in rice seeds.
Author Response
Thanks for your comments.
1. line 83, the authors should provide graphic genotype for IL3.1 and IL3.2;
Thanks. We have added the graphic genotype in Figure S1.
2. Page 2 line 92-94, there is no data to support this results, pls provide supporting data;
We have changed "OsFAD3" to “OsFAD3-1 or OsFAD3-2” in line 92-94.
3. Page 4, if possible, the authors could provide the expression data for OsFAD3-2 in rice seeds.
We have provided the expression data for OsFAD3-2 in rice embryo in Figure S1 and inserted a sentence of “The same pattern with a higher expression level of OsFAD3-2ZS97 than OsFAD3-2NIP was observed (Figure S1, C-D)” in L138-139.